# JEPA-REASONER: GENERATIVE LATENT SPACE REASONER

## ABSTRACT

While Joint-Embedding Predictive Architecture (JEPA) has emerged as a powerful architecture for learning rich latent representations, it fundamentally lacks generative abilities. Meanwhile, latent space reasoning attempts for Transformer models like COCONUT do improve performance, but they ultimately rely on token-by-token generation, which still accumulates compounding error and relies on context information to gain reasoning insights. To address these limitations, we propose JEPA-Reasoner, a novel JEPA model enhanced with generative ability that reasons in latent space. We augment it with a separate action-taker model, Talker, to produce human-readable sentences. Our approach demonstrates that decoupling latent space reasoning and token generation enables JEPA-Reasoner to produce mixed latent vectors that might lay the foundation for multi-threaded reasoning, while performing autoregressive generation with superior robustness to compounding error.

## 1 INTRODUCTION

The Joint-Embedding Predictive Architecture (JEPA) (Assran et al., 2023) has demonstrated strong performance in learning semantic world representations, exhibiting superior world understanding ability compared with traditional end-to-end generative models that work in pixel or token space. By predicting abstract representations in latent space, the JEPA architecture is able to filter irrelevant details and preserve essential information needed for prediction (Assran et al., 2023). Such architecture has been proven to be a viable approach for representation learning and foundation model development. Various JEPA implementations, including I-JEPA (Assran et al., 2023), V-JEPA2 (Assran et al., 2025), and M3-JEPA (Lei et al., 2025), have shown success across various modalities and downstream tasks.

However, JEPA models are inherently non-generative (Lei et al., 2025) because of their objective: filling missing information in the current state (Assran et al., 2023), rather than generating new content. Besides, the predictor often requires a detailed, predetermined target state or task instruction (e.g., V-JEPA 2-AC (Assran et al., 2025)), which is often unavailable in tasks demanding long-term planning and step-by-step reasoning. This feature limits the application of the broad knowledge in JEPA models to generative tasks.

Furthermore, while traditional token-level autoregressive models have sequential reasoning capabilities, their token-by-token generation process is prone to compounding errors. Even if the predicted probabilities of tokens are weighted and combined, LLMs still cannot go beyond single-threaded reasoners (Wu et al., 2025). Although several previous research have explored latent space reasoning for Transformer models, the end-to-end generation goal of a single coupled model limited the full potential of latent space reasoning, while also making training complex and inefficient (Hao et al., 2024).

To address these limitations, we propose JEPA-Reasoner, a novel decoupled architecture that utilizes separate models for reasoning and token generation. The reasoning model, JEPA-Reasoner, transforms the JEPA framework from a target-conditioned system into an autoregressive generative model. Operating entirely within the continuous, normalized latent space, JEPA-Reasoner focuses solely on latent space reasoning, offloading the token generation task to its action-taker module: Talker. In contrast to existing latent space reasoning solutions like COCONUT, the division of objective frees the reasoner from expression burden and enables continuous latent guidance (compared

with non-continuous latent guidance that relies on context information to retrieve latent reasoning results (Hao et al., 2024)) during the token generation process.

Our key insight is that performing pure reasoning in an abstract, continuous, and normalized latent space without token generation burden allows for the construction of high-level reasoning chains that carry rich semantic information throughout the autoregressive process while correctly ignoring irrelevant details or distracting information. Our empirical experiments show further improved abilities beyond that: by operating in the latent space, JEPA-Reasoner can maintain multiple hypotheses during the reasoning process simultaneously and mitigate the catastrophic error propagation associated with discrete token sampling.

## 2 RELATED WORK

**Joint-Embedding Predictive Architectures (JEPA).** JEPA (Assran et al., 2023) introduced a framework that makes predictions in representation space. It utilizes self-supervised training to learn latent states that are not directly human-readable. A predictor module was trained to predict the target state based on encoded inputs. Multiple variants of this architecture have extended the JEPA family to various modalities and downstream tasks (Assran et al., 2023). However, these models are non-generative. Attempts to make JEPA generative, such as D-JEPA (Chen et al., 2025), utilize the learned representations to condition diffusion models for data generation (e.g., text to images, text to audio) but still failed to enable sequential reasoning or planning within the JEPA framework itself. In contrast, our key innovation is to adapt the core JEPA objective for autoregressive latent-space generation.

**Latent Space Reasoning.** Previous work on latent space reasoning mainly focuses on looping hidden states, either through horizontal autoregression like COCONUT (Hao et al., 2024), or vertical recurrent depth scaling (Geiping et al., 2025). However, these paradigms utilize a single coupled model for both latent reasoning and token generation, ignores the mismatch between the two tasks: one requires high-level global planning, decision-making, and choice tracing, while the other requires local correctness in grammar and fluency. Besides, whether iterating over tokens or deepening the computation per token, both paradigms remain bound by causal masking, meaning early token generation cannot be guided by future reasoning states. Considering these limitations, our key innovation is to decouple latent space reasoning and token generation. By generating a complete latent chain first, we enable consistent latent guidance where all tokens are generated from the full reasoning trajectory. This produces an answer with higher quality that is less prone to error propagation. Additionally, the decoupled design enables efficient optimization with a single forward pass in latent space, unlike coupled models like COCONUT or Recurrent Depth transformers which require multiple synchronized passes or complex recurrent unrolling.

**Autoregressive Models and Robustness.** Modern Transformer models conduct token-level prediction in an autoregressive manner. While proven powerful on various tasks, this approach is known to suffer from compounding errors in long-horizon tasks. Techniques like Chain-of-Thought (Wei et al., 2022) improve reasoning by generating intermediate steps, but still operate at the token level. JEPA-Reasoner aims to improve robustness of autoregressive generation by moving the reasoning process into a continuous, abstract latent space, reducing the impact of localized errors.

## 3 MODEL ARCHITECTURE

JEPA-Reasoner decouples the reasoning process from output generation, making next-state predictions completely dependent on previously generated, semantic-rich, lossless latent states. The architecture consists of:

- **JEPA-Reasoner:** Generate sequential latent space reasoning chains independently.

- **Talker:** Translates the latent states into tokens. Note that the Talker is not able to make predictions. Its task is to reconstruct tokens based entirely on the latent output from JEPA-Reasoner.

## 3.1 JEPA-REASONER

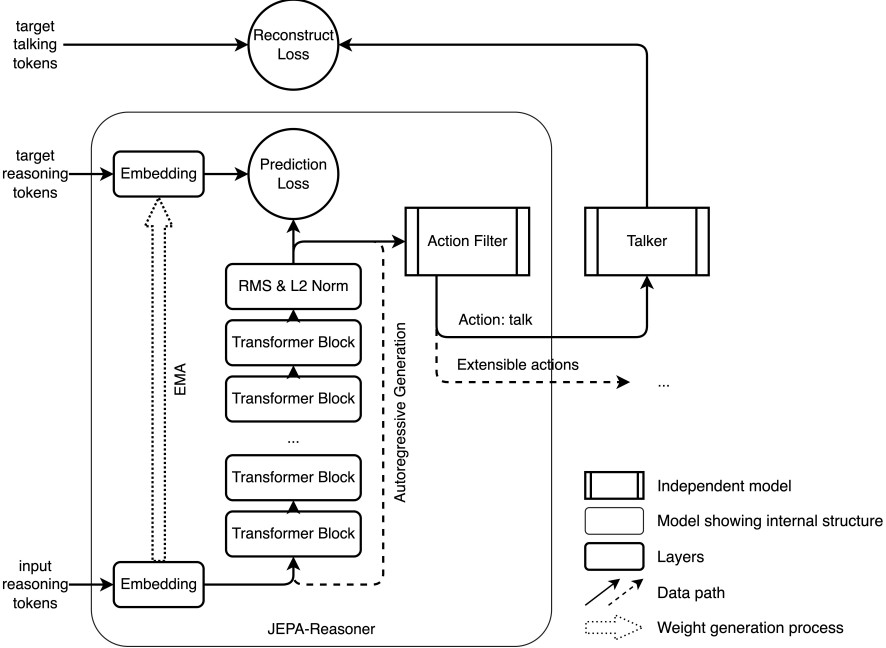

Figure 1: Architecture of JEPA-Reasoner and its action taker (Talker). The Reasoner consists of an embedding layer as token encoder and Transformer blocks as predictor. The embedding layer for input tokens always uses the latest weights, while the weight of embedding layer for target tokens is the exponential moving average of the historical weights of input embedding layer.

**Model Components.** JEPA-Reasoner follows the JEPA philosophy, containing an embedding layer as a textual token encoder and modified Transformer blocks for the predictor, since Transformer has proven its strong ability in sequence modeling. After the modified Transformer blocks, we applied a hybrid normalization layer (RMS and L2 normalization). We utilize L2 normalization to prevent exploding magnitude caused by residual connection. In the modified Transformer block, we apply a non-learnable QK-Norm (Dehghani et al., 2023) to make it more numerically stable.

**Latent Space Generation.** Unlike traditional JEPA models, in which the predictor is aimed at filling missing information in the current state (Assran et al., 2023), the predictor of JEPA-Reasoner generates the *next* latent matrix representing the subsequent reasoning steps. Crucially, this generated latent matrix is not projected into vocabulary probabilities via an LM head. Instead, it is normalized by the hybrid normalization layer to the unit hypersphere and looped back as the input of the first Transformer block for the next round of autoregressive generation in the latent space.

**Training Objective and Target Encoder.** The model is trained to predict latent representations provided by a target encoder. Following standard JEPA methodology, the target encoder weights are an exponential moving average (EMA) of the data encoder weights, providing stable and rich training targets. Given the normalized nature of our latent space, we use scaled cosine similarity loss computed entirely in latent space, ensuring the predictor learns consistent feature representations and dynamics (refer to Section 4.2 for more detail).

## 3.2 ACTION-TAKER MODEL

In this experiment, there is only one action-taker model: Talker. The Talker model is a standard Transformer-based model trained independently. We designed two variants of Talker: Mono-Talker and Dual-Talker. Detailed information about the components of the two Talkers is shown in Table 1. Mono-Talker does not have an embedding layer or encoders, it only has decoders. Mono-Talker is

designed for reconstruction tasks that do not require context information, receiving latent vectors from JEPA-Reasoner and constructs the complete token sequence in one forward pass. Dual-Talker is designed for context-aware reconstruction, usually necessary in natural language tasks. It has an embedding layer, encoders and decoders. The embedding layer is used for encoding previously determined outputs of JEPA-Reasoner that contain contextual information, while the encoder blocks receive latent vectors from JEPA-Reasoner as input. The decoders generate tokens autoregressively conditioned on previous tokens, with continuous latent guidance from the output of encoders. However, Dual-Talker was trained for reconstruction rather than generation, as our ablation study (Appendix C) showed that Talker is critically dependent on the Reasoner's output. During training, the JEPA-Reasoner is frozen. Talker receives the sequence of latent vectors and is trained to reconstruct the corresponding token sequence using standard cross-entropy loss.

| | Embedding Layer | Standard Encoder | Standard Decoder | LM Head |
|---|---|---|---|---|
| Mono-Talker | No | No | Yes | Yes |
| Dual-Talker | Yes | Yes | Yes | Yes |

Table 1: Components of Mono-Talker and Dual-Talker

### 3.3 ACTION FILTER

In scenarios requiring interaction with different modalities or tools, the JEPA-Reasoner can generate specific "action latent vectors" that signal the need to invoke a specific action module. The Action Filter detects these markers and routes the subsequent latent vectors to the appropriate module. While this detection could be handled by a trained MLP classifier, our experiments focus solely on text generation, utilizing hard-coded action filters based on the training data structure to simplify evaluation.

## 4 TRAINING PROCEDURE

The training process of JEPA-Reasoner consists of two main phases. The first stage is pretraining that teaches basic knowledge (e.g., grammar and commonsense) to the model. The second stage is self-supervised training (SST), which adapts the model to perform consistent latent space reasoning.

### 4.1 PRETRAINING

We apply established Transformer training methods to provide the model with basic knowledge and language understanding capabilities.

**Objective and Methodology.** The model is trained as a standard decoder-only Transformer on the next-token prediction task in a teacher-forcing way. We employ tied word embeddings which shares the weight of embedding layer with a temporary LM head. The LM head is only used in pretraining and is removed after pretraining is finished. The L2 normalization layer is disabled in the pretraining phase to enable simple and direct reuse of current Transformer training recipes. Considering that tied word embedding encourages $W_{Embed} \cdot W_{Embed}^T = I$ and $v_{pred} \cdot v_{embed} = \|v_{pred}\| \cdot \|v_{embed}\| \cdot \cos(\theta)$, the tied-embedding approach will indirectly encourage angular alignment between predicted vectors and embedding vectors, which facilitates the subsequent transition from token-level to latent-level prediction.

### 4.2 SELF-SUPERVISED TRAINING

The SST phase adapts the pretrained model for making consistent predictions in the continuous latent space. Since the model is fully transforming into a latent space reasoner instead of a token generator, the ability to produce correct logits no longer matters. Considering this, we apply similar self-supervised training as Meta's JEPA series (Assran et al., 2023) in this stage. Without the need for autoregressively generating final token outputs to compute a loss, self-supervised training enables efficient parallel training compared with COCONUT (Hao et al., 2024).

**Objective and Methodology.**    The temporary LM head in the pretraining stage was discarded and L2 normalization layer was restored. The model is now optimized to predict the latent representation of the next sequence segment, with a consistent dimensional semantics as what the embedding layer produces. We switch to scaled cosine distance loss, aligning with the L2 normalization strategy used to ensure stability during autoregressive looping and focusing the learning on angular differences:

$$\mathcal{L}(\theta, \theta') = k - k \cdot \cos(h_{\text{pred}}(\theta), h_{\text{target}}(\theta')) \tag{1}$$

where $k$ is the scalar, $h_{\text{pred}}$ is the predicted latent vector by the Reasoner (parameters $\theta$), and $h_{\text{target}}$ is the target latent vector from the EMA encoder (parameters $\theta'$). In our empirical tests, we find that normal cosine distance loss failed to support enough optimization when the loss is small. We tested a series of $k$ values, and chose $k = 4$ in our experiments (Refer to Appendix D for more detail).

**Target Generation.**    The weight updating method of the target embedding layer is different from input embedding layer. The input embedding layer always applies the latest weights, while the target embedding layer utilizes exponential moving average to generate its weights from the historical weights of input embedding layer. We applied a high momentum value of 0.98 to prevent rank collapse in the embedding layer while ensuring enough space to adjust for angular alignment.

## 5    LATENT SPACE PROPERTIES

We analyze the property of JEPA-Reasoner's latent representation on two synthetic tasks designed to probe specific capabilities in controlled environments: mixed latent vector generation via a tree-search problem, and robustness to error propagation via a Context-Free Grammar (CFG) generation task.

### 5.1    CONTINUOUS REPRESENTATION OF UNCERTAINTY

Within a reasoning process, JEPA-Reasoner is able to produce mixed latent vectors that are not limited to the discrete representations in the embedding layers. The mixed latent vectors approximate a linear combination of more than one vocabulary latent (latent vectors that correspond to individual vocabulary tokens). To systematically examine this behavior, we trained a smaller JEPA-Reasoner (42M) to search routes from the root to specific leaves in a binary tree.

#### 5.1.1    DATA PREPARATION

Training data consists of randomly generated binary trees with depth limited to 4. Each tree node was represented by a character with a unique token, making up the vocabulary along with other special tokens. In the generation process, we randomly pick node names to prevent the model from memorizing relationships based on names. Refer to Appendix A for an example.

#### 5.1.2    MODEL CONFIGURATION

The JEPA-Reasoner model and Mono-Talker model were built with specifications stated in Table 2. We chose the combination of JEPA-Reasoner with Mono-Talker because this task does not require context-aware reconstruction.

|  | Latent Dim. | Attention Dim. | FFN Dim. | Head Count | Decoder Count |
|---|---|---|---|---|---|
| JEPA-Reasoner | 384 | 768 | 1536 | 16 | 18 |
| Mono-Talker | 384 | 768 | 1536 | 8 | 6 |

Table 2: Model configurations in tree-search experiment

#### 5.1.3    TRAINING

The pretraining and SST process are completely the same as stated in Section 4, except for loss masking. In the pretraining stage, loss was computed on all positions, while in SST, loss was only computed on the positions that define the desired route. When training the Talker model, we only

passed latent vectors that describe the route to Talker to ensure it had no access to the tree structure or the target leaf, which guaranteed the Talker could not solve the task on its own.

### 5.1.4 RESULTS AND CONCLUSIONS

The final combination of the JEPA-Reasoner and Mono-Talker models achieved 99.87% accuracy (exact match) in searching routes from the tree root to specific leaves. Given the restricted context window of the Mono-Talker model, we could confirm that only JEPA-Reasoner was responsible for reasoning. Based on this result, we examine the generated latent vectors to probe the reasoning behavior of JEPA-Reasoner.

We calculated the distance from the predicted latent vector to the plane spanned by any two vocabulary vectors and sorted them from closest to farthest. In the sorted list, the plane spanned by latent vectors of sibling nodes frequently exhibits lower distances to the predicted latent vector, with an average ranking of top 1.72% in the ordered list. Also, we figured out all coefficient sets, $\alpha$ and $\beta$, that satisfy $\alpha \cdot l_0 + \beta \cdot l_1 = l_{proj}$, where $l_0$ and $l_0$ are latent vectors of sibling nodes and $l_{proj}$ is the projection of the predicted latent vector on the spanned plane. After comparing $\alpha$ and $\beta$, we find that for 99.89% of the times, the latent vector of the node on the correct route contributes more than the other sibling node. This discovery demonstrated that JEPA-Reasoner could make correct choices without completely discarding the other information that contains potentially correct choices. According to the previous COCONUT study (Hao et al., 2024), this behavior might lay the foundation for breadth-first multi-threaded reasoning.

### 5.2 ROBUSTNESS TO ERROR PERTURBATION

In the following sections, we demonstrate that decoupling reasoning chain generation from token production enables superior robustness under noisy conditions. While coupled models must simultaneously maintain reasoning coherence and produce correct tokens, our decoupled approach allows the reasoning model to focus solely on maintaining logical consistency in latent space, exhibiting better generation quality. This section contains two experiments that focuses on two different sources of errors: token level error and latent space noise.

### 5.2.1 EXPERIMENT METHODS

**Robustness Test for Token Level Error**   To evaluate the robustness of the decoupled model on token-level errors in the input sequence, we randomly replace 0% to 30% ground truth tokens in the input sequence with incorrect tokens. We compare the performance of JEPA-Reasoner and traditional Transformer models on multi-step completion tasks[1] using the exact match metric.

**Robustness Test for Latent Space Error**   To evaluate the robustness of JEPA-Reasoner model on perturbations in latent space, we compare the performance of JEPA-Reasoner and the coupled continuous reasoning model COCONUT on multi-step completion tasks. In this experiment, we let the COCONUT model autoregressively generate 4 latent vectors first, followed by 4 tokens. For JEPA-Reasoner we simply let it generate 8 latent vectors and use the Talker module to reconstruct 8 tokens. For both models, we add Gaussian noise to the generated latent vector at each step with $\mu = 0$ and $\sigma$ ranging from 0% to 15% of the maximum value in the model's output. Accuracy is calculated across the last 4 tokens with the exact match metric.

### 5.2.2 DATA PREPARATION

Considering that the compounding error caused by different faulty tokens differs significantly, it is difficult to quantitatively analyze the model's behavior under token-level errors (e.g., replacing keywords in the sentence will decrease the quality more considerably than replacing a word that functions as a connector). We followed previous work by Allen-Zhu & Li (2023) and utilized Context-Free Grammar (CFG) for both experiments to create a controllable experiment setting.

---

[1] All scores obtained in these two tests are by testing the model across 5248 samples randomly chosen from the test dataset containing 100000 samples to minimize the bias introduced by randomness.

Our custom CFG production rule features three terminal symbols with rule lengths of 3 or 4. With this rule, we generated long (approximately 600 to 700 symbols) and complex sequences that require non-trivial work to solve. The complexity of the grammar ensures that high accuracy relies on learning the underlying structure of the CFG sequence rather than memorizing specific sequences (refer to Appendix B.1 for full CFG specifications and production methods).

### 5.2.3 Model Configurations and Training Methods

We denote the vanilla Transformer model as $T$, the COCONUT-style coupled latent space reasoning model as $C$, and the decoupled model as $R$ (both JEPA-Reasoner and Talker are included). We made variants of these models in three scales: $small$, $middle$, and $large$. Table 3 shows more detailed model configurations:

|  | $R_{large}$ | $T/C_{large}$ | $R_{middle}$ | $T/C_{middle}$ | $R_{small}$ | $T/C_{small}$ |
|---|---|---|---|---|---|---|
| Total Parameters | 315M | 338M | 209M | 229M | 132M | 157M |
| Latent Dimension | 960 | 960 | 960 | 960 | 960 | 960 |
| Attention Dimension | 960 | 960 | 960 | 960 | 960 | 960 |
| FFN Dimension | 3840 | 3840 | 3840 | 3840 | 3840 | 3840 |
| Head Count | 16 | 16 | 16 | 16 | 16 | 16 |
| Talker Block Count | 4 + 4 | – | 2 + 2 | – | 2 + 2 | – |
| Reasoner Block Count | 16 | – | 12 | – | 6 | – |
| Transformer Block Count | – | 24 | – | 16 | – | 10 |
| Total Blocks | 24 | 24 | 16 | 16 | 10 | 10 |

Table 3: Model Configurations for CFG Task. Talker Block Count format (E+D) refers to Encoder and Decoder blocks in the Dual-Talker model. COCONUT models and Transformer models are put in the same column, since they share the same architecture.

We apply identical hyperparameters (with learning rate of $1 \times 10^{-4}$, effective batch size of 128, and context length of 1024) to train all models until their loss stabilizes, then checkpoints of best performance were chosen as the representative. We first pretrain the Transformer models on CFG data using cross-entropy loss in token space. Since the pretraining methods of Transformer, CO-CONUT, and JEPA-Reasoner are identical, subsequent trainings are based on the same pretraining checkpoints[2]. We conduct posttraining to obtain Transformer models: $T_{small}$, $T_{middle}$, and $T_{large}$. COCONUT models are trained to first predict 4 hidden states, then generate 4 tokens. Cross-entropy loss is computed between the output logits and the target sequence, excluding the hidden state positions. We follow the training method mentioned in Section 4 to train the JEPA-Reasoner and Dual Talker models until their loss stabilizes.

### 5.2.4 Results and Conclusions

Our robustness evaluation demonstrates the advantages of the decoupled architecture. In the token level error experiment, JEPA-Reasoner showed less performance degradation across different model scales when facing input noise during multi-step CFG completion tasks (Figure 2). Large variant of JEPA-Reasoner also exhibits higher performance across different magnitudes of Gaussian noise in the latent space error experiment (Table 4), providing more empirical evidence for its robustness advantage.

These results demonstrate that JEPA-Reasoner has the potential to address the limitations of existing paradigms in Section 2. By operating in a normalized latent space and offloading token generation to the Talker module, subsequent reasoning outputs do not condition on previous decisions, thus mitigating error accumulation in the autoregressive process, enabling more robust sequential generation under noisy conditions.

---

[2]Due to architectural differences, we initialize JEPA-Reasoner models using only the first $N$ blocks from the pretrained Transformer, where $N$ matches the JEPA-Reasoner's block count.

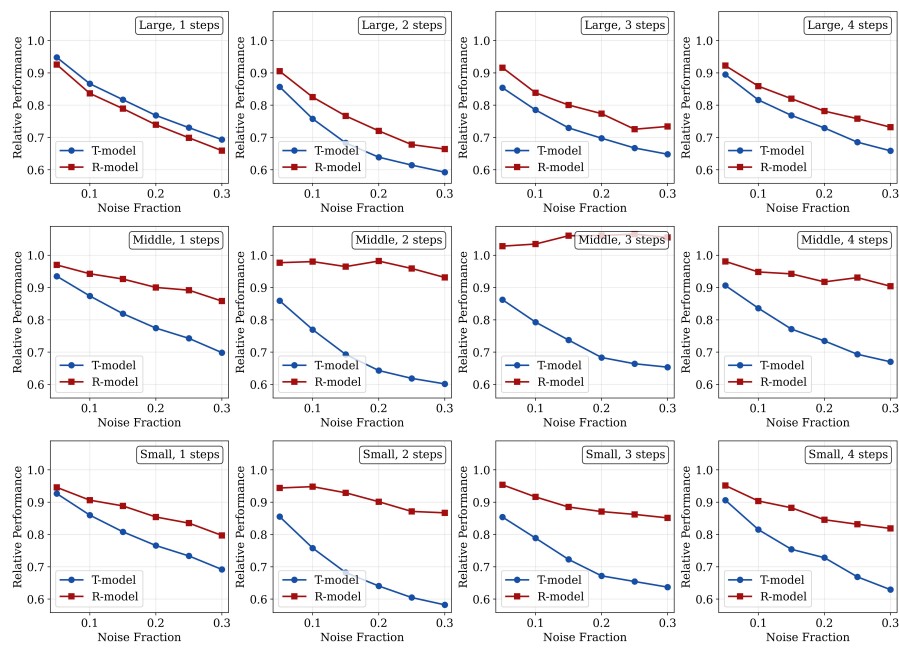

Figure 2: Relative performance of coupled token-level ($T$) models and decoupled ($R$) models across configurations.

|  | $\sigma = 0.0$ | $\sigma = 0.05 \times \max{(h_t)}$ | $\sigma = 0.10 \times \max{(h_t)}$ | $\sigma = 0.15 \times \max{(h_t)}$ |
|---|---|---|---|---|
| $R_{large}$ | 0.4588 | 0.4681 | 0.4643 | 0.4468 |
| $C_{large}$ | 0.3740 | 0.3688 | 0.3650 | 0.3629 |
| $R_{middle}$ | 0.2973 | 0.3039 | 0.3049 | 0.3023 |
| $C_{middle}$ | 0.3792 | 0.3802 | 0.3798 | 0.3761 |
| $R_{small}$ | 0.3342 | 0.3315 | 0.3318 | 0.3312 |
| $C_{small}$ | 0.3864 | 0.3773 | 0.3677 | 0.3550 |

Table 4: Performance of JEPA-Reasoner $R$ and COCONUT $C$ under different noise levels.

## 6 REAL WORLD EVALUATION

We trained a 694M JEPA-Reasoner and a 198M Mono-Talker on natural language using the training steps stated in section 4. We analyze the performance in two ways:

1. Performance difference between pretrained model (token level prediction) and JEPA-Reasoner (latent level prediction), paired with its Mono-Talker model.

2. Performance difference between JEPA-Reasoner (paired with Mono-Talker) and other latent or non-latent reasoning models.

### 6.1 PERFORMANCE GAIN AFTER LATENT REASONING ADAPTATION

We evaluated both the base Transformer model and JEPA-Reasoner on GSM8k (Cobbe et al., 2021). Table 5 shows the benchmark score under both 5-shot and 8-shot settings.

| Model | Accuracy (%) | |
|---|---|---|
|  | 5-shot | 8-shot |
| Base Transformer | 20.7 | 20.8 |
| JEPA-Reasoner + Talker | 37.1 | 48.2 |

Table 5: GSM8k benchmarks of base Transformer and JEPA-Reasoner

**Performance Gain.** Comparing to base Transformer, JEPA-Reasoner improves performance by 79.2% (5-shot) and 131.7% (8-shot) respectively, demonstrating that adapting to a decoupled latent space reasoning paradigm significantly increases the accuracy of expressing internally leaned knowledge with explicitly stated token sequences. Considering that it takes 300k steps for pretraining but only 13k steps for SST, while using the same dataset, it shows great evidence that most of the performance gain came from the adaptation to latent reasoning, rather than SST steps. Also, the scaled cosine similarity loss used in SST encourages smooth latent representation transition rather than logical correctness, which further support that minimal new knowledge was learned during SST.

**In Context Learning.** Crucially, while the Base Transformer's performance stagnates between 5-shot and 8-shot settings (improving only by 0.1%), the JEPA-Reasoner utilizes the additional examples to nearly double its accuracy. This scaling behavior indicates that the decoupled latent space reasoning architecture effectively overcomes the reasoning plateau, which was widely observed in small-scale token-base models, allowing for more robust logical deduction that is less constrained by surface-level token statistics.

## 6.2 Performance Comparison with Other Reasoning Models

We compare JEPA-Reasoner against standard Transformer models and other reasoning models. Table 6 shows the GSM8k benchmark performance of our JEPA-Reasoner model (paired with Talker) compared to other models of similar size or performance.

| Reasoning Paradigm | Example | Model Size | 8-shot Accuracy (%) |
|---|---|---|---|
| Standard | Gemma 3 | 4B | 38.4 |
| CoT | Llama 3.2 | 1B | 44.4 |
| CoT | Qwen 3 | 0.6B | 42.5 |
| Recurrent Depth | Huginn-0125 | 3.5B | 42.1 |
| **Ours** | **JEPA-Reasoner** | **0.9B** | **48.2** |

Table 6: GSM8k benchmark comparison between JEPA-Reason and other reasoning models. Note that we could not find 8-shot GSM8k benchmark results for COCONUT models. Since COCONUT models sacrifice performance (compared to CoT) as a trade-off for efficiency (Hao et al., 2024), we did not include them in the performance comparison.

**Analysis of Results.** As illustrated in Table 6, JEPA-Reasoner demonstrates superior performance on the GSM8k benchmark compared to both standard transformer baselines and dedicated CoT models. With a parameter count of only 0.9B, our model achieves an 8-shot score of 48.2%, outperforming Llama 3.2 (1B) CoT baseline by 3.8 percentage points. Most notably, JEPA-Reasoner significantly surpasses larger models that rely on standard reasoning paradigms. Despite being approximately $4\times$ smaller than Gemma 3 (4B), our model exhibits a performance gain of nearly 10%. While the Recurrent Depth model (Huginn-0125) offers a strong baseline at 42.1%, it requires nearly four times the parameter count to achieve results that are still 6.1% lower than JEPA-Reasoner. Consequently, these results serves as strong empirical evidence of JEPA-Reasoner's capability to handle complex natural language reasoning tasks, effectively applying mathematical logic to solve problems.

## 7 Theoretical Advantage of Decoupled Architecture

The robustness of our decoupled architecture stems from its ability to decouple the high-level reasoning process from low-level token generation. We can formalize this advantage by analyzing the probabilistic assumptions and information flow within the coupled and decoupled paradigms.

### 7.1 Model Dynamics and Probabilistic Factorization

Let $R = (r_1, r_2, \ldots, r_T)$ be the sequence of latent reasoning states and $X = (\hat{x}_1, \hat{x}_2, \ldots, \hat{x}_T)$ be the sequence of generated tokens.

**Classical Transformer Model.** A standard Transformer model $\mathcal{M}_t$, implicitly defines a joint probability distribution that is factorized sequentially. The generation of the state and token at step $t$ depends on both the state and the sampled token from step $t - 1$:

$$P(R, X) = \prod_{t=1}^{T} P(r_t, \hat{x}_t | r_{t-1}, \hat{x}_{t-1})$$

In this formulation, the distribution for the next reasoning state $r_t$ is directly conditioned on the previously sampled token $\hat{x}_{t-1}$. Consequently, a sampling error at step $t - 1$ (i.e., $\hat{x}_{t-1} \neq x^*_{t-1}$) introduces a persistent error into the reasoning state trajectory. This error corrupts the foundation for all subsequent reasoning and generation steps, leading to compounding error.

**COCONUT Model.** The COCONUT model $\mathcal{M}_c$ is a coupled model with latent generation ability. It generates latent reasoning tokens before producing final tokens (Hao et al., 2024). Limited by the coupled architecture, latent vectors and tokens are arranged in the same sequence $Z = (z_1, z_2, \ldots, z_N)$, where each element $z_t$ can be either a continuous latent vector $r_t$ or a discrete token $\hat{x}_t$. Despite the different output types, the model follows the same autoregressive principle: every new element is conditioned on all prior elements.

$$P(Z) = \prod_{t=1}^{N} P(z_t | z_{<t})$$

Let's consider a generation length of $T_1$ for latent reasoning, followed by $T_2$ tokens. The process unfolds as follows:

- For the latent reasoning steps ($t = 1, \ldots, T_1$), the model generates $z_t = r_t$, conditioning on the previous latent vectors $z_{<t} = (r_1, \ldots, r_{t-1})$.
- For the token generation steps ($t = T_1 + 1, \ldots, T_1 + T_2$), the model generates $z_t = \hat{x}_{t-T_1}$, conditioning on the full history of all preceding latent vectors and tokens, $z_{<t} = (r_1, \ldots, r_{T_1}, \hat{x}_1, \ldots, \hat{x}_{t-T_1-1})$.

The unified sequence is the model's critical limitation. Suppose the model has finished generating its reasoning chain $(r_1, \ldots, r_{T_1})$ and the first error appeared at the $n^{th}$ token, $\hat{x}_n \neq x^*_n$. For the very next step, $t = T_1 + n + 1$, the model make prediction based on the history $(r_1, \ldots, r_{T_1}, \hat{x}_1, \ldots, \hat{x}_n)$. The erroneous token $\hat{x}_n$ is now an immutable part of the model's context, corrupting every subsequent decision. The error propagation is direct and unavoidable because reasoning and generation are inextricably linked in the same autoregressive sequence.

**Decoupled Model $\mathcal{M}_d$:** In contrast, our JEPA-Reasoner architecture imposes a structural constraint on the generative process, yielding a more robust factorization where the reasoning chain is generated independently of the token sampling:

$$P(R, X) = P(R) \cdot P(X|R) = \left(\prod_{t=1}^{T} P(r_t | r_{t-1})\right) \cdot \left(\prod_{t=1}^{T} P(\hat{x}_t | R, \hat{x}_{1:t-1})\right)$$

This factorization reveals two key theoretical advantages.

1. **Error Containment:** The reasoning trajectory's probability, $P(R)$, is independent of the token generation process $P(X|R)$. An error in sampling a token $\hat{x}_{t-1}$ has *no mathematical pathway* to influence the reasoning trajectory $R$. The high-level plan remains intact and stable. Furthermore, the normalization of reasoning vectors $r_t$ to the unit hypersphere ensures this trajectory is inherently bounded, preventing error amplification within the reasoning dynamics itself.

2. **Mechanism for Recovery:** At every step $t$, the token generator $P(\hat{x}_t|\cdot)$ is conditioned on the *entire, lossless* reasoning chain $R$. This provides a strong, stable signal that allows the Talker to potentially recover from a local token error in its own history ($\hat{x}_{1:t-1}$), an effect empirically validated in our ablation study (Appendix C).

This inherent error containment and recovery mechanism explains the superior robustness observed in our CFG experiments (Section 5.2).

## 8 SUMMARY

We introduce JEPA-Reasoner, a novel architecture that decouples latent space reasoning from token generation. Our approach enables continuous latent reasoning guidance while mitigating step-by-step error propagation. Efficient parallel training was also made possible without sacrificing latent reasoning performance compared with COCONUT. Our experiments on synthetic tasks suggest that by decoupling the high-level latent space reasoning process from low-level token generation, JEPA-Reasoner produced promising potential for multi-threaded reasoning and exhibited enhanced robustness to input noise and error accumulation when generating structured sequences.

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

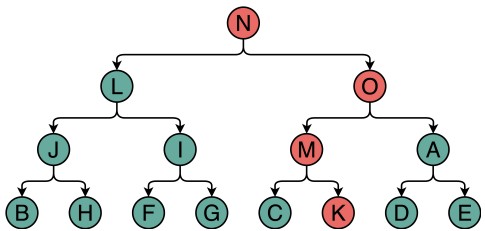

Figure 3: Visualization of the tree structure in the given example.

Chünhung Wu, Jinliang Lu, Zixuan Ren, Gangqiang Hu, Zhi Wu, Dai Dai, and Hua Wu. Llms are single-threaded reasoners: Demystifying the working mechanism of soft thinking. *arXiv preprint arXiv:2508.03440*, 2025. URL `https://arxiv.org/abs/2508.03440`.

## A  DATA FOR TREE SEARCH EXPERIMENT

The following is an example of data used in the tree-search experiment:

```
NL,NO,LJ,LI,OA,OM,JB,JH,IG,IF,AD,A
E,MK,MC[ROOT]N[TARGET]K[ROUTE]NOMK
```

Visualization of the example can be seen in Figure 3. In the sequence, each character pair represents a parent-child node pair, with the former one being the parent node and the later one being the child. All pairs are separated by a comma. The searching task is specified after the tree-structure definition, with special token [ROOT] indicating the tree root, [TARGET] indicating which leaf to search for, and [ROUTE] states the correct searching route. All characters, comma, [ROOT], [TARGET] and [ROUTE] have a corresponding token, making up the whole vocabulary for the model along with the padding token and the end-of-sentence token.

## B  FURTHER DETAILS FOR CFG EXPERIMENTS

### B.1  CFG RULES AND SAMPLE

CFG can hierarchically produce highly structured expressions by replacing non-terminal symbols at each level with next-level symbols following a production rule, as shown in Figure 4. A sequence of terminal symbols is considered to be valid if it can be transformed back to the root symbol with dynamic programming and the given production rule. The recursive structure and local ambiguity of CFG sequences enable them to model the rich and recursive structure in languages, including grammar and logic. We designed our own CFG following the method used by Allen-Zhu & Li (2023). The production rule used in our experiments is a five-level CFG production rule set featuring three terminal symbols with 3 or 4 rule lengths at each level, which typically generates long (typically 600 to 700 symbols per sample) and locally ambiguous sequences. A visualization of the production rule used in this experiment can be seen in Figure 5.

Since even a 5-level CFG production rule that allows each non-terminal symbol to produce 2 to 3 symbols in the next level (simpler than our 5-level production rule that allows each non-terminal symbol to produce 3 to 4 symbols in the next level) is capable of producing more than $4 \times 10^8$ distinctive sequences, we conclude that the models in the CFG experiments does not rely on memorizing possible sequences during training to achieve high accuracy on completion tasks.

Previous research (Allen-Zhu & Li, 2023) shows that Transformer blocks can encode the structure of CFG rules within parameters. We assume that a robust model should be able to recognize the high-level structure of the input sequence, thus ignoring faulty tokens in the input. Since each high-level element in our CFG sequence produces 3 to 4 tokens, the model should be able to maintain relatively stable performance across at least 4 generation steps.

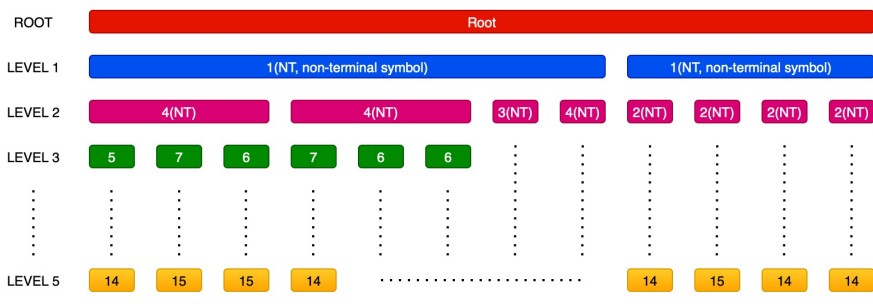

Figure 4: A picture demonstrating how CFG sequence is generated. It involves replacing non-terminal symbols at each level with symbols from the next level with a given rule.

```
LEVEL 1        LEVEL 2        LEVEL 3          LEVEL 4            LEVEL 5

1->4 4 4 3     2->6 5 6       5->9 9 10        8->13 12 11        11->15 16 16
1->2 2 2 2     2->7 6 5 5     5->10 10 8       8->13 12 11        11->14 16 14
               3->7 5 7 6     6->8 9 10 8      9->11 13 12        11->15 16 16 14
               3->6 6 7       6->9 8 9 10      9->12 12 13 13     11->15 15 16 14
               3->6 5 7 6     7->9 9 8 10      9->13 12 12        12->15 15 14 14
               3->6 6 6 5     7->9 9 10 9      10->12 13 12 12    12->15 14 14 15
               4->5 7 6                        10->13 12 12       13->15 16 14
               4->6 5 6 5                      10->11 13 12       13->14 15 14 15
               4->6 6 7 6                                         13->15 16 16
               4->7 6 6
```

Figure 5: CFG production rule used to generate training and test samples in Section 5.2. This rule gives sequence lengths ranging from about 600 symbols to 700 symbols.

## B.2 A SAMPLE CFG SEQUENCE

We demonstrate a sample CFG sequence from the training dataset: `15 16 16 14......14 16 14 (666 terminal symbols in total)`. It consists of three kinds of terminal symbols.

## B.3 DETAILED EXPERIMENT RESULTS

We provide detailed results for the accuracy of different models and configurations on different noise levels here in Table 7. Note that the accuracy is the total correct symbols generated divided by the total symbols needed in the generation task.

## B.4 PARAMETER EFFICIENCY COMPARISON

As demonstrated in Appendix C, the Talker does not participate in the construction of the reasoning chain, letting JEPA-Reasoner handle all reasoning alone, which means the effective parameter count for reasoning is actually smaller than the total parameter count. Also, the total number of parameters of the reasoner-talker pair is always a bit smaller than the corresponding Transformer or COCONUT counterpart because it is not possible to make it exactly the same in good practice (e.g., it is not ideal to use an odd number as embedding dimension size). Considering this, JEPA-Reasoner is at a disadvantage in model size, which may result in the drop of absolute performance seen in small

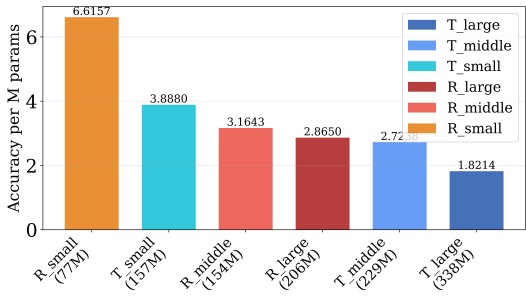

Figure 6: The parameter efficiency of JEPA-Reasoner and Transformer models at different scales.

| Noise | Model | T models | | | | R models | | | |
|---|---|---|---|---|---|---|---|---|---|
| | | Step 1 | Step 2 | Step 3 | Step 4 | Step 1 | Step 2 | Step 3 | Step 4 |
| 0.00 | Small(abs) | 92.4 | 92.0 | 62.3 | 67.5 | 78.0 | 51.5 | 44.6 | 52.1 |
| | Middle(abs) | 92.9 | 92.8 | 63.3 | 67.9 | 73.1 | 47.9 | 35.4 | 46.3 |
| | Large(abs) | 92.3 | 91.8 | 61.9 | 68.6 | 91.9 | 74.1 | 57.2 | 66.5 |
| 0.00 | Small(rel) | 100.0 | 100.0 | 100.0 | 100.0 | 100.0 | 100.0 | 100.0 | 100.0 |
| | Middle(rel) | 100.0 | 100.0 | 100.0 | 100.0 | 100.0 | 100.0 | 100.0 | 100.0 |
| | Large(rel) | 100.0 | 100.0 | 100.0 | 100.0 | 100.0 | 100.0 | 100.0 | 100.0 |
| 0.05 | Small(abs) | 85.6 | 78.6 | 53.2 | 61.1 | 73.8 | 48.6 | 42.5 | 49.6 |
| | Middle(abs) | 86.8 | 79.8 | 54.5 | 61.5 | 70.9 | 46.8 | 36.4 | 45.4 |
| | Large(abs) | 87.5 | 78.7 | 52.8 | 61.4 | 85.1 | 67.0 | 52.4 | 61.3 |
| 0.05 | Small(rel) | 92.6 | 85.4 | 85.4 | 90.5 | 94.6 | 94.4 | 95.3 | 95.2 |
| | Middle(rel) | 93.4 | 86.0 | 86.1 | 90.6 | 97.0 | 97.7 | 102.8 | 98.1 |
| | Large(rel) | 94.8 | 85.7 | 85.3 | 89.5 | 92.6 | 90.4 | 91.6 | 92.2 |
| 0.10 | Small(abs) | 79.5 | 69.7 | 49.1 | 55.0 | 70.7 | 48.8 | 40.8 | 47.1 |
| | Middle(abs) | 81.2 | 71.5 | 50.2 | 56.8 | 68.9 | 46.9 | 36.6 | 43.9 |
| | Large(abs) | 80.0 | 69.6 | 48.6 | 56.0 | 76.9 | 61.1 | 48.0 | 57.1 |
| 0.10 | Small(rel) | 86.0 | 75.8 | 78.8 | 81.5 | 90.6 | 94.8 | 91.5 | 90.4 |
| | Middle(rel) | 87.4 | 77.0 | 79.3 | 83.7 | 94.3 | 97.9 | 103.4 | 94.8 |
| | Large(rel) | 86.7 | 75.8 | 78.5 | 81.6 | 83.7 | 82.5 | 83.9 | 85.9 |
| 0.15 | Small(abs) | 74.7 | 62.7 | 45.0 | 50.9 | 69.3 | 47.8 | 39.5 | 46.0 |
| | Middle(abs) | 76.1 | 64.3 | 46.7 | 52.4 | 67.7 | 46.2 | 37.5 | 43.6 |
| | Large(abs) | 75.4 | 62.7 | 45.1 | 52.7 | 72.6 | 56.8 | 45.8 | 54.5 |
| 0.15 | Small(rel) | 80.8 | 68.2 | 72.2 | 75.4 | 88.8 | 92.8 | 88.6 | 88.3 |
| | Middle(rel) | 81.9 | 69.3 | 73.8 | 77.2 | 92.6 | 96.5 | 105.9 | 94.2 |
| | Large(rel) | 81.7 | 68.3 | 72.9 | 76.8 | 79.0 | 76.7 | 80.1 | 82.0 |
| 0.20 | Small(abs) | 70.8 | 58.9 | 41.9 | 49.1 | 66.7 | 46.4 | 38.8 | 44.1 |
| | Middle(abs) | 71.9 | 59.7 | 43.2 | 49.9 | 65.8 | 47.0 | 37.5 | 42.4 |
| | Large(abs) | 70.9 | 58.7 | 43.2 | 50.0 | 68.0 | 53.4 | 44.3 | 51.9 |
| 0.20 | Small(rel) | 76.6 | 64.0 | 67.3 | 72.7 | 85.5 | 90.1 | 87.0 | 84.6 |
| | Middle(rel) | 77.4 | 64.3 | 68.2 | 73.5 | 90.0 | 98.1 | 105.9 | 91.6 |
| | Large(rel) | 76.8 | 63.9 | 69.8 | 72.9 | 74.0 | 72.1 | 77.4 | 78.0 |
| 0.25 | Small(abs) | 67.8 | 55.6 | 40.8 | 45.1 | 65.2 | 44.9 | 38.4 | 43.4 |
| | Middle(abs) | 69.0 | 57.5 | 42.0 | 47.1 | 65.2 | 45.9 | 37.7 | 43.0 |
| | Large(abs) | 67.4 | 56.4 | 41.3 | 47.0 | 64.3 | 50.2 | 41.5 | 50.4 |
| 0.25 | Small(rel) | 73.4 | 60.4 | 65.5 | 66.8 | 83.6 | 87.2 | 86.1 | 83.3 |
| | Middle(rel) | 74.3 | 62.0 | 66.4 | 69.4 | 89.2 | 95.8 | 106.5 | 92.9 |
| | Large(rel) | 73.0 | 61.4 | 66.7 | 68.5 | 70.0 | 67.7 | 72.6 | 75.8 |
| 0.30 | Small(abs) | 63.9 | 53.6 | 39.7 | 42.5 | 62.2 | 44.6 | 38.0 | 42.7 |
| | Middle(abs) | 64.9 | 55.9 | 41.3 | 45.5 | 62.7 | 44.6 | 37.3 | 41.8 |
| | Large(abs) | 64.0 | 54.4 | 40.1 | 45.2 | 60.6 | 49.2 | 42.0 | 48.6 |
| 0.30 | Small(rel) | 69.2 | 58.3 | 63.7 | 63.0 | 79.7 | 86.6 | 85.2 | 82.0 |
| | Middle(rel) | 69.9 | 60.2 | 65.2 | 67.0 | 85.8 | 93.1 | 105.4 | 90.3 |
| | Large(rel) | 69.3 | 59.3 | 64.8 | 65.9 | 65.9 | 66.4 | 73.4 | 73.1 |

Table 7: Robustness Comparison: Accuracy (%) across different noise fractions and generation steps (e.g., "Step $k$" in the table means the average accuracy across $k$ generation steps). "abs" is absolute performance, while "rel" is the model's relative accuracy compared with clean data input.

and medium-sized models. To make the comparison of absolute performance fair, we calculated the parameter efficiency, using the formula $\frac{\hat{s}}{p}$, where $\hat{s}$ is the average absolute performance and $p$ is the parameter count of JEPA-Reasoner, and gained Figure 6:

In the parameter efficiency comparison, decoupled JEPA-Reasoners consistently show advantages compared with their coupled Transformer counterparts, proving that for every million parameters, the reasoning component (JEPA-Reasoner) in the decoupled architecture gains more performance compared with traditional coupled Transformer models.

## C ABLATION STUDY OF TALKER MODEL

We conduct an ablation study to verify two critical properties of our decoupled architecture:

1. **Reasoning Dominance:** The reasoning process is strictly driven by the JEPA-Reasoner's latent trajectory, preventing the Talker from bypassing the architecture to perform independent inference.

2. **Linguistic Capability:** Talker module acts as an effective "Language Interface," capable of translating abstract latent vectors into grammatically correct and semantically coherent natural language.

We test this by corrupting the output of Reasoner in different ways. Our empirical experiments show strong evidence that the Talker cannot reason on its own and serves primarily as a readout mechanism for the Reasoner's planning.

### C.1 EXPERIMENT SETUP

Using the training method mentioned in Section 4, we use the JEPA-Reasoner model initialized with Transformer blocks trained on C4 and Wikitext(Merity et al., 2016) dataset in this experiment setup to produce a human-readable result. We conducted controlled experiments using two sample inputs from the training dataset to evaluate the dependency of the Talker model on the Reasoner's output:

The first sample is: "Francis Bacon was an English philosopher and statesman who served as Attorney General and Lord Chancellor of England under King James I. Bacon argued for the importance of natural philosophy, guided by the scientific, his works remained influential". This sample is used as JEPA-Reasoner and Talker's input unless mentioned otherwise. The second sample is "Jean-Paul Sartre was a French philosopher, political activist, biographer, and literary critic. Sartre was one of the key figures in the philosophy of existentialism (and phenomenology).", which is used in the "Semantic Mismatch" experiment

We systematically corrupted different components of the input to isolate the contribution of each part:

- Baseline: Normal operation with clean Reasoner output
- Random String Replacement: Replace Reasoner output with a random string
- Initial Token Corruption: Keep Reasoner output clean, but replace Talker's initial input string (decoder input) with a random string.
- Gaussian Noise: Replace Reasoner output with Gaussian noise ($\mu = 0$, $\sigma = 1$)
- Semantic Mismatch: Use Reasoner output from a different sentence

### C.2 RESULTS

Table 8 presents the results of our ablation experiments. The Talker's initial input tokens were the first 10 tokens of the Francis Bacon sample (*Francis Bacon was an English philosopher and statesman who* in natural language) across most experiments unless mentioned otherwise.

**Latent Dependency (Proof of No Independent Reasoning).** When the Reasoner output is replaced with Gaussian noise, the Talker produces incoherent output. This demonstrates that the Talker cannot generate meaningful content without proper latent representations.

**Robustness to Token Errors (Proof of Linguistic Capability).** When only the initial input tokens are corrupted, the Talker can still produce largely coherent content, guided by the clean Reasoner output, although some localized errors occur (e.g., "Richardson" appears). This suggests the Reasoner's latent representations carry the primary semantic information.

**Semantic Fidelity (The Decoupling Verification).** When using the Reasoner output from the Jean-Paul Sartre sample as Dual-Talker's input latent. Although the first 10 tokens from the Francis Bacon sample are used as the initial input tokens of decoder blocks in Dual-Talker, it rapidly shifts

| Condition | Talker Output |
|---|---|
| **Baseline** | Bacon was an English philosopher and statesman who served as Attorney General and Lord Chancellor of England under King James I. Bacon argued for the importance of natural philosophy, guided by the scientific; his works remained influential |
| **Random String as Reasoner Input** | Francis Bacon was an English philosopher and statesman whoTGvIUujapaUYDIUbIyuTviuiYtvU |
| **Random Initial Tokens for Talker** | SsDXicoundfdfiyx served as Attorney General and Lord Chancellor of England under King James I. Richardson argued for the importance of natural philosophy, guided by the scientific, his works remained influential |
| **Gaussian Noise in Reasoner Output** | Francis Bacon was an English philosopher and statesman who 0 not deathsardumn technicalipt'tputalase2006 (ightsringst 0q virtually lowesteral 1 new date 2007 0 results; |
| **Semantic Mismatch** | Francis Bacon was an English philosopher and statesman who, political activist, biographer, and literary critic. Zartre was one of the key figures in the philosophy of existentialism (and phenomenology). |

Table 8: Ablation Study Results: Talker Model Output Under Different Input Corruptions.

to generating the Jean-Paul Sartre content. This provides strong evidence that the Talker genuinely utilizes the semantic content encoded by the JEPA-Reasoner. Meanwhile, the fact that the generated output remains grammatically fluid despite this conflict proves the Talker successfully handles the linguistic realization of the Reasoner's abstract concepts.

## D  K VALUE IN SCALED COSINE DISTANCE LOSS

We tested $k$ from 1 to 6. All these $k$ values could produce a basic SST outcome that exhibits reasoning behaviors stated in all previous sections. With a careful tuning of $k$, we observed a stable improvement in the tree-search problem as shown in Figure 7:

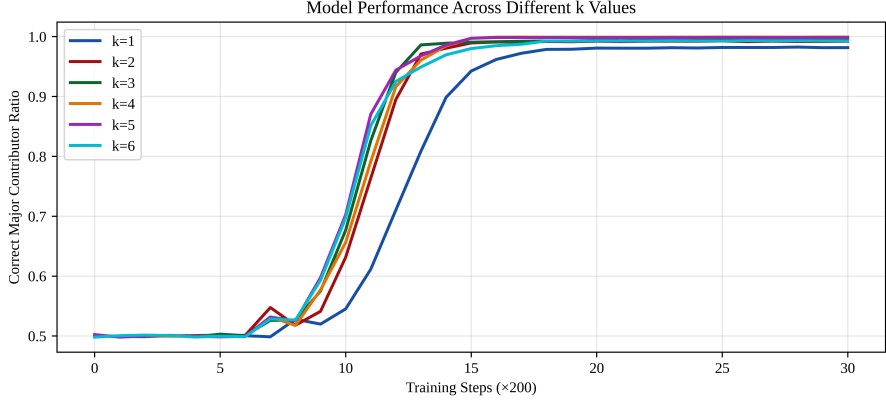

Figure 7: Changes of correct major contributor rate with training steps. Zoom in to see details.

We choose the correct major contributor (the correct next-step latent vector plays the most important role in current-step mixed latent vectors) rate as the metric since it directly relates to the correctness of future predictions. Considering that when $k = 4$, the model gains the highest correct major

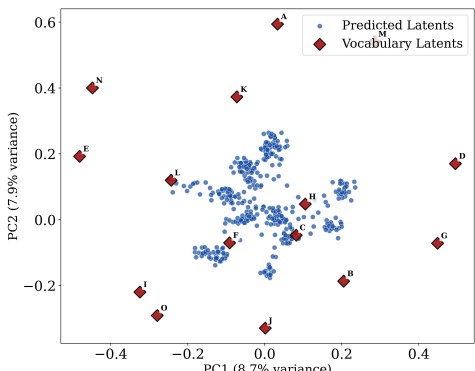

Figure 8: PCA analysis of latent representations of different tree leaves in the tree search experiment.

contributor rate (zoom in to distinguish the line of $k = 4$ from the line of $k = 5$), we choose to continue our experiment with $k = 4$.

## E  VISUALIZATION OF MIXED LATENT VECTORS

To visualize that JEPA-Reasoner can produce mixed latent vectors, we gathered the embedding vectors and model predictions from our tree-search experiments in this section. We extracted embedding vectors from distinct tree leaves alongside the model's output latent representations after one forward pass. Principal Component Analysis (PCA) was applied to the collected embeddings and model predictions, and the visualization focuses on the first two principal components (PC1 and PC2).

As demonstrated in the Figure 8, predicted latent vectors (blue points in the figure) form a continuous cloud within the space spanned by discrete vocabulary embeddings (red diamond shapes). This distribution supports the experiment results that they are the linear combinations of vocabulary embeddings. Also, the predicted vectors do not converge to singular vocabulary points, providing empirical evidence for the hypothesis that JEPA-Reasoner is capable of maintaining information from multiple possible choices rather than committing to a single answer.

