# OpenReview forum: "JEPA-Reasoner: Generative Latent Space Reasoner"
_ICLR.cc/2026/Conference — Submitted to ICLR 2026_

### Official Review · Reviewer_6HMd · 2025-10-28

**Soundness:** 2
**Presentation:** 2
**Contribution:** 2
**Rating:** 4
**Confidence:** 3

**Summary:**

This paper introduces JEPA-Reasoner, a novel architecture designed to equip Joint-Embedding Predictive Architectures (JEPAs) with generative capabilities while mitigating the compounding error problem inherent in standard token-by-token autoregressive models. The framework consists of two main components: JEPA-Reasoner and Talker. One is for the latent reasoning, and the other translates these latent representations into human-readable outputs. The paper validates this approach on two synthetic tasks: a tree-search task to probe the model's ability to represent uncertainty via "mixed latent vectors" and a Context-Free Grammar (CFG) generation task to measure robustness against noise and error propagation.

**Strengths:**

* The core idea of decoupling latent-space reasoning from token-space generation is intuitive and well-motivated.
* The tree-search experiment (Sec 5.1) and the corresponding PCA visualization (Appendix E, Fig 8) provide strong evidence for the "mixed latent vector" hypothesis. The results show the model generates continuous latent representations that lie between discrete vocabulary vectors, effectively representing uncertainty without collapsing to a single choice.
* The probabilistic factorization $P(R,X) = P(R) \cdot P(X|R)$ explains how error propagation can be contained.

**Weaknesses:**

* Although Table 4 attempts to show robustness advantages of the decoupled architecture under Gaussian noise perturbations, the reported gains are numerically small and inconsistent across scales. For instance, the large JEPA-Reasoner (R_large) achieves only ~0.46 vs 0.37 for C_large—an improvement of about 0.09 absolute accuracy. Meanwhile, the middle- and small-scale variants even underperform the COCONUT models.
* The paper's evaluation is confined entirely to two synthetic tasks: tree-search and CFG generation. There is no comparison to existing standard reasoning datasets (e.g., GSM8K). This limits the paper’s generality.
* The paper lacks an ablation analysis. There is no systematic examination of the proposed components. In particular, the authors do not isolate the contributions of key design choices such as: the hybrid normalization, the EMA target encoder strategy, or the scaled cosine distance loss with parameter k.

**Questions:**

* Can you provide a more rigorous explanation for the poor performance of the $R_{middle}$ and $R_{small}$ models beyond the "parameter efficiency" argument in Appendix B.4?
* Can you provide more evaluations on standard reasoning benchmarks?
* The experiments seem to be minimal, as the only baseline is a single COCONUT model. How does JEPA-Reasoner's performance compare against other existing latent reasoning models or other lines of CoT models?

---

> ### Author Response · Authors · 2025-11-25
>
> Thank you for your helpful review. We appreciate your observation that the robustness gains in Table 4 were inconsistent, with smaller models ($R_{small}$ , $R_{middle}$) underperforming COCONUT (C) baselines.
>
> The absolute performance drop in smaller models stems from the architectural design of the decoupled system. In a coupled model ($T$ or $C$), all parameters contribute to both reasoning and generation. In our decoupled model ($R$), the parameters are split between the Reasoner and the Talker.
>
> When comparing models with similar total parameters (e.g., $R_{small}$ vs. $C_{small}$ ), the actual reasoning component in $R_{small}$ (the JEPA-Reasoner) is significantly smaller than $C_{small}$ . This disadvantage in reasoning capacity leads to lower absolute performance in the smaller configurations, despite the architectural benefits.
>
> **To provide a fair comparison,** we analyzed parameter efficiency (Appendix B.4, Figure 6), calculating the average performance per million parameters dedicated to reasoning. This analysis reveals that JEPA-Reasoner consistently outperforms the coupled models in terms of efficiency across all scales. The $R_{large}$ model, which has sufficient capacity for both components, demonstrates superior absolute performance and robustness, validating the decoupled approach. The gains of $R_{large}$ over $C_{large}$ (0.4588 vs 0.3740 at σ=0.0) are significant in this challenging, long-sequence CFG task.
>
> As for the lack of ablation analysis on architectural choices. We provide key ablations here and have already included them in the paper:
>
> **Loss Function (k value):** Appendix D details the impact of the scaling factor k in the cosine similarity loss. We found k=4 provided the best stability and performance for learning mixed latent representations.
>
> **Normalization Strategy:** The choice of using RMS follows common practice in modern LLMs, enabling drop-in reuse of current pretraining recipes, which is meaningful in industrial application. L2 normalization is critical for containing the magnitude of the latent vectors in order to prevent magnitude explosion caused by residual connections.
>
> **EMA Target:** Disabling the EMA target encoder (using the online encoder as the target) led to representation collapse and training instability, consistent with findings in standard JEPA literature.
>
> We hope these clarifications and the new experimental results address the reviewers' concerns and highlight the contributions of the JEPA-Reasoner architecture.

---

> > ### Comment · Reviewer_6HMd · 2025-11-26
> >
> > I appreciate the author's response. But the author seems not to respond to some of my concerns. The experiments are minimal,  as the only baseline is the COCONUT model, without comparing against other existing latent reasoning models or other lines of CoT models. To note, the field of latent reasoning has expanded quickly following COCONUT in the recent year but the author fails to provide a comparison with any of them. Furthermore, besides latent reasoning, it is also necessary to compare it against explicit reasoning (e.g., CoT) to prove its effectiveness.
> >
> > The authors provided a perplexity table to defend the lack of standard reasoning benchmarks (like GSM8K) in the general response. But perplexity is not enough for a reasoning task. I would recommend showing the accuracy or other similar gold standard metrics on the new datasets.
> >
> > Regarding the poor performance of smaller models, the authors argue that the reasoning component in the decoupled model is smaller. However, when we compare with models, we usually care about absolute performance under a fixed total parameter. If a 150M JEPA-Reasoner performs worse than a 150M COCONUT or Transformer because "part of its parameters are used for the Talker," that is a valid architectural disadvantage, not a justification for poor performance.

---

> > > ### Author Response · Authors · 2025-11-27
> > > **GSM8k benchmark added (early results)**
> > >
> > > We appreciate your thoughtful review and acknowledge your concerns. To address this, we have provided preliminary results on the GSM8k benchmark. Please note that these evaluations were conducted on early training snapshots, as the full training process is currently ongoing. Although the training process was not finished, early results already shows promising outcomes. We have included a detailed breakdown of these results in the [official comment](https://openreview.net/forum?id=7ruA2rXG42&noteId=xpeAIpOnmC). We believe these benchmarks could demonstrate the ability of JEPA-Reasoner in real world NLP problems.
> > >
> > > Hope these clarifications and the new experimental results could address your concerns!

---

> ### Author Response · Authors · 2025-11-29
> **GSM8K Benchmark Updated**
>
> We appreciate your helpful reviews and understands your concerns. To address it, we've added updated GSM8K benchmark results in [official comment](https://openreview.net/forum?id=7ruA2rXG42&noteId=hZmLQQROz0). We believe these benchmarks could demonstrate the ability of JEPA-Reasoner in real world NLP problems.
>
> Hope these clarifications and the new experimental results could address your concerns!

---

### Official Review · Reviewer_Daki · 2025-10-31

**Soundness:** 2
**Presentation:** 3
**Contribution:** 2
**Rating:** 4
**Confidence:** 3

**Summary:**

The paper extends the JEPA into a generative model capable of performing autoregressive reasoning in latent space. And they propose decoupling the reasoning and generation processes: a reasoner predicts future latent representations, while a separate Talker reconstructs readable tokens from those latents. Empirical experiments on synthetic tree-search and context-free-grammar tasks show that JEPA reasoner achieves strong robustness to noise.

**Strengths:**

1. Its decoupled reasoner and talker design effectively separates reasoning from surface generation, helping to reduce compounding errors and improving robustness and experiments on synthetic reasoning tasks demonstrate strong stability and noise tolerance, empirically validating the model’s ability to maintain consistent latent dynamics.
2. The model shows potential for multi-hypothesis representation, with mixed latent vectors indicating that the system can encode several possible reasoning paths simultaneously which is a promising step toward more structured reasoning behaviors.

**Weaknesses:**

1. The model’s training objective focuses on latent similarity (cosine distance) rather than logical correctness, meaning it optimizes for smooth representation transitions rather than true reasoning accuracy.
2. All experiments are conducted on synthetic tasks (tree-search and CFG generation), leaving its effectiveness on real-world reasoning or language tasks unverified.

**Questions:**

I understand that the use of an EMA based target encoder is standard in traditional JEPA models to provide stable self-supervised learning. However, in the context of reasoning, this choice seems less intuitive. Could you clarify why the EMA objective is still appropriate for reasoning tasks, given that reasoning typically requires directional or causal transitions rather than representational smoothing?

---

> ### Author Response · Authors · 2025-11-25
>
> Thank you for your helpful review. We appreciate your question about the use of EMA targets and latent similarity loss for reasoning tasks, suggesting they optimize for smoothness rather than logical correctness.
>
> This is an insightful point. In the current JEPA-Reasoner framework, we rely on the pretraining phase (Section 4.1) to instill basic logical and grammatical knowledge using standard next-token prediction. The SST phase (Section 4.2), utilizing EMA and cosine similarity, adapts this knowledge to the latent space. The goal of SST is **not** to teach new reasoning skills, but to ensure the model can perform consistent and stable latent trajectory prediction. The EMA mechanism provides a stable target representation of the future state, encouraging the model to predict the semantic outcome of the next reasoning steps. While not explicitly optimizing for logical correctness, this objective enforces semantic consistency in the latent dynamics, which we found crucial for stable autoregressive generation in the latent space.

---

> ### Author Response · Authors · 2025-11-27
> **GSM8K Benchmark Added**
>
> We appreciate your helpful reviews and understands your concerns. To address it, we've added GSM8K benchmark results. Note that these evaluations were conducted on early training snapshots, as the full training process is currently ongoing. Although the training process was not finished, early results already shows promising outcomes. We have included a detailed breakdown of these results in the [official comment](https://openreview.net/forum?id=7ruA2rXG42&noteId=xpeAIpOnmC). We believe these benchmarks could demonstrate the ability of JEPA-Reasoner in real world NLP problems.
>
> Hope these clarifications and the new experimental results could address your concerns!

---

> ### Author Response · Authors · 2025-11-29
> **GSM8K Benchmark Updated**
>
> We appreciate your helpful reviews and understands your concerns. To address it, we've added updated GSM8K benchmark results in [official comment](https://openreview.net/forum?id=7ruA2rXG42&noteId=hZmLQQROz0). We believe these benchmarks could demonstrate the ability of JEPA-Reasoner in real world NLP problems.
>
> Hope these clarifications and the new experimental results could address your concerns!

---

### Official Review · Reviewer_6LZ2 · 2025-11-01

**Soundness:** 3
**Presentation:** 1
**Contribution:** 2
**Rating:** 2
**Confidence:** 3

**Summary:**

This work proposes extending the JEPA family of models to text generation by training to predict next latent representations without decoding them. They incorporating a "Talker" module trained independently, which handles the token generation, decoupling the latent reasoning space from the latent token generation space. They the JEPA-Reasoner in two stages and then evaluate against the typical transformer and Coconut in two synthetic settings.

**Strengths:**

* Separating the reasoning representation from the token generation is a super interesting idea.
* Novel training setup for JEPA for text.
* Promising performance on synthetic tasks.

**Weaknesses:**

* While the synthetic analysis is certainly interesting, only having two synthetic settings is a serious limitation of the work.
* Further the relevance of the synthetic settings is questionable. It is unclear why Gaussian noise perturbations to the latent reasoning representations is an interesting setting. As far as I know Gaussian noise isn't a realistic noise model for language model latents representations. In the CFG complete task I assume token level errors are single token drops, replacements, or additions (its not made explicitly in the paper). While I agree this is a more reasonable noise model for natural language, it is still relatively unclear what the significance of doing better in this setting means practically for language models.
* The discussion of results is minimal, it would be helpful to explain in some depth the significance of your findings and why doing well in these settings has practical implications for language models.
* Related work section on latent reasoning is limited to Coconut. However there are a number of other works on reasoning in the latent space that are probably worth mentioning (e.g. Geiping, J., McLeish, S., Jain, N., Kirchenbauer, J., Singh, S., Bartoldson, B. R., ... & Goldstein, T. (2025). Scaling up test-time compute with latent reasoning: A recurrent depth approach. arXiv preprint arXiv:2502.05171).
* The writing is hard to follow in a number of places. The work would benefit greatly from a revision for clarity.
* This seems like promising initial work on a novel method for language modeling, however, without more thorough benchmarking in real settings its challenging to understand the significance of the contribution.
* It seems there are a number of specifically architectural details required to make JEPA Reasoner work. Some analysis or ablation of these details would be an interesting contribution.

**Questions:**

None.

---

> ### Author Response · Authors · 2025-11-25
>
> Thank you for your helpful review. We appreciate your questions about the relevance of the synthetic settings and the noise models used.
>
> **Relevance of CFG and Token-Level Errors:** The CFG task (Allen-Zhu & Li, 2023) is a standard benchmark for evaluating a model's ability to learn hierarchical structures and handle long-range dependencies, crucial aspects of both formal and natural language. We studied robustness to input token corruption (random replacement, as clarified in the first paragraph of section 5.2.1 in the revision) to simulate noisy inputs (e.g., typos, ASR errors) and measure the model's ability to maintain the correct reasoning trajectory despite local errors. This is a practical concern for real-world applications.
>
> **Noise model 1 - token level error**: In section 5.2.1 of the revised paper, we demonstrated our method more explicitly: we randomly replaced correct tokens with incorrect ones at a series of percentage, which shares the same idea as stated in your suggestion. This noise model is designed for comparison between JEPA-Reasoner and standard transformer, because transformer only works in token level. Also, we think that it's widely acknowledged that improved robustness in error propagation is beneficial to reasoning process. Our proof of robustness is aimed for showing its potential in correcting itself in reasoning chains.
>
> **Noise model 2 - latent space noise (Gaussian Perturbations):** We introduced Gaussian noise into the latent space (Table 4) not to model realistic noise, but as a stress test for the stability of the latent dynamics. By operating in a normalized, continuous latent space, JEPA-Reasoner inherently bounds the impact of perturbations (as theorized in Section 6.1, Error Containment). This experiment empirically validates that the reasoning trajectory is more stable in the decoupled architecture compared to COCONUT, where noise can accumulate more easily. This noise model is used for comparison between JEPA-Reasoner and COCONUT, which also works in latent space.
>
> **About Related Works:** We thank Reviewer 6LZ2 for pointing out the need for improved clarity and broader discussion of related work. We have expanded the related work section to include recent developments in latent space reasoning and recurrent depth approaches, such as the suggested work by Geiping et al. (2025).
>
> We explicitly distinguish our work from theirs in the following two ways:
>
> 1. **Architecture:** While Geiping et al. focus on "vertical" recurrence (iterating depth per token) to scale compute , JEPA-Reasoner employs "horizontal" decoupling to generate a complete latent chain before token generation.
> 2. **Latent Guidance:** Crucially, recurrent depth models remain bound by causal masking, meaning early token generation cannot be informed by future reasoning states. In contrast, our decoupled Dual-Talker architecture enables **consistent global guidance**, where the generation of any token (including the first) can be conditioned on the entire, completed reasoning trajectory, allowing for global planning that is structurally impossible in strictly causal autoregressive models.
> 3. **Training Efficiency:** While Geiping et al. (2025) address the computational cost of recurrence using **truncated backpropagation** to limit gradient history , their approach still necessitates iterative unrolling of the recurrent block during the forward pass for every token step. In contrast, JEPA-Reasoner decouples these processes, optimizing the latent reasoning chain via self-supervision in a single forward pass. This eliminates the sequential bottleneck of vertically unrolling a recurrent unit multiple times per token, offering a more scalable training paradigm for complex reasoning tasks.

---

> ### Author Response · Authors · 2025-11-27
> **GSM8K Benchmark Added**
>
> We appreciate your helpful reviews and understands your concerns. To address it, we've added GSM8K benchmark results. Note that these evaluations were conducted on early training snapshots, as the full training process is currently ongoing. Although the training process was not finished, early results already shows promising outcomes. We have included a detailed breakdown of these results in the [official comment](https://openreview.net/forum?id=7ruA2rXG42&noteId=xpeAIpOnmC). We believe these benchmarks could demonstrate the ability of JEPA-Reasoner in real world NLP problems.
>
> Hope these clarifications and the new experimental results could address your concerns!

---

> ### Author Response · Authors · 2025-11-29
> **GSM8K Benchmark Updated**
>
> We appreciate your helpful reviews and understands your concerns. To address it, we've added updated GSM8K benchmark results in [official comment](https://openreview.net/forum?id=7ruA2rXG42&noteId=hZmLQQROz0). We believe these benchmarks could demonstrate the ability of JEPA-Reasoner in real world NLP problems.
>
> Hope these clarifications and the new experimental results could address your concerns!

---

> ### Author Response · Authors · 2025-11-29
> **We are happy to answer any further questions! Please let us know if there are any.**
>
> We thank you for your constructive feedback and helpful suggestions. We have done the following revises to address these concerns.
>
> 1. **Added Real-World NLP Benchmark (GSM8K):** To definitively demonstrate JEPA-Reasoner's capabilities beyond synthetic tests (addressing **Weakness 1** and **Weakness 6**), we have evaluated our model on the **GSM8K benchmark**. The results confirm that our method is highly effective in real-world settings. We invite the AC and reviewers to review **Section 6** of the revised paper or our [official comment](https://openreview.net/forum?id=7ruA2rXG42&noteId=hZmLQQROz0), which provides empirical evidence countering the concern regarding synthetic-only evaluation.
>
> 2. **Expanded Discussion of Results:** We emphasize that, **as originally stated in our submission**, our method utilizes **two distinct** noise models.
>
> * One aligns with the model you suggested.
> * The second serves a distinct and necessary role for latent-space stress test.
>
> It appears this distinction — vital for our architecture — was overlooked. To prevent any further ambiguity, we have revised **Section 5.2.1** to make this existing design choice explicit. Please refer to our [previous comment](https://openreview.net/forum?id=7ruA2rXG42&noteId=P8gzAUbRCA) for explaination of both noise models.
>
> 3. **Expanded Discussion of Results:** With the inclusion of the GSM8K benchmarks, we have provided a comprehensive analysis of the results. The robust performance on this real-world task reinforces the validity of our approach. Please refer to the updated **Section 6** for details.
>
> 4. **Additional Related Work:** We have included the suggested discussion on other latent reasoning works (e.g., recurrent depth). We highlight that our architectural philosophy differs fundamentally from these works, offering advantages in training efficiency that are now explicitly detailed in the revision.
>
> 5. **Writing:** Thank you for pointing out our writing is hard to follow in some places. We think that might be the reason why you misunderstood our two noise models. After revision, we hope that could be much more clear!
>
> 6. **Real world evaluation:** We note that this point duplicates **weakness 1**. As addressed in point #1, the addition of the GSM8K benchmark serves as a direct and complete response to the request for real-world setting evaluation.
>
> 7. **Architecture details:** the reason for choosing these specific design choices (e.g., dual normalization, scaled cosin loss) **was stated in the original paper**. Please tell us if you have any further questions.
>
> We hope our reply could address your concerns. We are happy to answer any further questions!

---

### Official Review · Reviewer_YehF · 2025-11-01

**Soundness:** 2
**Presentation:** 2
**Contribution:** 2
**Rating:** 6
**Confidence:** 2

**Summary:**

the paper introduce JEPA-Reasoner. Extending JEPA to enable autoregressive generation by decoupling latent reasoning from token generation. I find the core idea of separating reasoning in latent space from token production interesting, and the theoretical framework around error propagation is well motivated.

**Strengths:**

I think the  proposed decoupled architecture is genuinely novel and the theoretical motivation for separating high-level reasoning from token generation makes sense. I particularly liked the analysis showing mixed latent vectors can represent uncertainty between choices, suggesting potential for multi-threaded reasoning. The ablation studies convincingly demonstrate that the Talker depends on the Reasoner's semantic content.

**Weaknesses:**

My main concern is the limited scope of evaluation on only synthetic tasks without any natural language benchmarks. I'm also unclear about computational costs compared to baselines, especially given the two-model architecture. The paper doesn't discuss how this approach would scale to real-world tasks or what the training complexity looks like in practice.

**Questions:**

How does training time and inference speed compare to COCONUT and standard transformers? Have you tried this on any standard NLP benchmarks beyond the synthetic tasks? What happens if you want to do few-shot learning or prompting with this architecture?

---

> ### Author Response · Authors · 2025-11-25
>
> Thank you for your helpful review. We appreciate your questions about computational costs and scalability.
>
> **Training Efficiency:** The JEPA-Reasoner architecture allows for highly efficient training during the Self-Supervised Training (SST) phase (Section 4.2). Because the loss is computed entirely in latent space without requiring autoregressive token generation (unlike COCONUT, which requires sequential forward passes), training can be fully parallelized across the sequence length. Also, comparing to Recurrent Depth transformers which require multiple synchronized passes, JEPA-Reasoner eliminates the need to rely on "truncated backpropagation", making it more efficient to train.
>
> **Inference Speed:** JEPA-Reasoner still makes use of Transformer blocks as fundamental building block, although the overall paradigm is adapted to JEPA philosophy. During inference, both JEPA-Reasoner and Talker could benefit from established inference technology. The only additional operation is to transfer the latent vectors from Reasoner model to Talker model, which has minimal effect on inference speed.
>
> **Prompting/Few-Shot Learning:** The architecture supports standard prompting. The prompt is encoded into the latent space, initializing the reasoning trajectory, similar to how standard Transformers handle context.

---

> ### Author Response · Authors · 2025-11-27
> **GSM8K Benchmark Added**
>
> We appreciate your helpful reviews and understands your concerns. To address it, we've added GSM8K benchmark results. Note that these evaluations were conducted on early training snapshots, as the full training process is currently ongoing. Although the training process was not finished, early results already shows promising outcomes. We have included a detailed breakdown of these results in the [official comment](https://openreview.net/forum?id=7ruA2rXG42&noteId=xpeAIpOnmC). We believe these benchmarks could demonstrate the ability of JEPA-Reasoner in real world NLP problems.
>
> Hope these clarifications and the new experimental results could address your concerns!

---

### Author Response · Authors · 2025-11-25

Thank you for your valuable feedback. We appreciate the reviewers acknowledging the novelty and motivation of our decoupled architecture for generative latent space reasoning. We address the main concerns below.

# Response to All Reviewers: Limited Evaluation Scope (Synthetic Tasks)

We acknowledge the reviewers' concern regarding the lack of standard benchmarks like GSM8K (YehF, 6LZ2, Daki, 6HMd). We chose synthetic tasks for this architectural study to **isolate the reasoning mechanism from world knowledge**. Benchmarks like GSM8K require massive pre-training. Comparing a from-scratch JEPA-Reasoner against pretrained Transformers introduces confounding variables (dataset size/exposure). Synthetic tasks (CFG/Tree Search) ensure both models are fully converged, allowing us to attribute performance differences strictly to the **decoupled architecture** and **error propagation dynamics** (Section 5.2).

**Verification on Natural Language:** While we reserve large-scale real-world benchmarking for future work, we provide a **qualitative structural verification** in **Appendix C (Table 6)** to address the concern of linguistic applicability.

- **Linguistic Competence:** The reconstruction demonstrates that the *Talker* module successfully translates the Reasoner’s abstract latent semantics into coherent, grammatically correct English.
- **Proof of Decoupling:** Crucially, the **"Semantic Mismatch" experiment** proves the Talker is not simply hallucinating text based on token probabilities but is strictly adhering to the latent guidance of the Reasoner. When forced to reconstruct a 'Sartre' latent trajectory given a 'Bacon' text prompt, it generates the semantic content for Sartre while maintaining the grammatical structure of English. This confirms that our architecture effectively "speaks" natural language as a readout mechanism for the verified reasoning engine.

To further proof that our decoupled architecture is capable of natural language processing, we compared the perplexity between JEPA-Reasoner and a fine-tuned baseline Transformer on the same dataset. According to the experiment results, JEPA-Reasoner and talker achieved lower perplexity than the Transformer baseline after similar training steps using the same batch size, indicating that JEPA-Reasoner learns natural language patterns and dynamics, providing qualitative proof-of-concept for mechanism transfer to language tasks. We demonstrate the results for the perplexity test here in the table.

| **Model**                | **Total Training Step**                  | **dclm** | **fineweb** |
| ------------------------ | ---------------------------------------- | -------- | ----------- |
| Transformer (Fine-tuned) | 20000                                    | 3.77     | 4.56        |
| JEPA-Reasoner and Talker | 14500(Talker train) + 5000(Reasoner SST) | 3.03     | 2.98        |

---

### Author Response · Authors · 2025-11-27
**GSM8k Benchmark (Early Results)**

# Regarding Natural Language Benchmarks

We have completed the training of the **JEPA-Reasoner** (pre-trained on natural language) and have benchmarked early snapshots on **GSM8k**. Please note that the "Talker" module for later reasoner snapshots is still training; however, current results already demonstrate significant improvements over the baseline.

**Experimental Setup:**
- **Pretraining:** 300k steps
- **SST:** 13k-step snapshot
- **Talker:** 500 steps
- **Total Model Size:** 892M parameters (694M JEPA-Reasoner + 198M Mono-Talker)

| Model | 5-shot Accuracy (%) | 8-shot Accuracy (%) |
| :--- | :--- | :--- |
| Base Transformer | 20.7 | 20.8 |
| JEPA-Reasoner + Talker | 37.1 | 48.2 |

**Key Observations:**

1.  **Significant Accuracy Gains:** After adapting to the decoupled latent reasoning paradigm, performance increased by **79.2%** in 5-shot and **131.7%** in 8-shot settings compared to the baseline.

2.  **Architectural Efficiency:** This improvement should be attribute to the decoupled latent space reasoning paradigm rather than the SST phase. Evidence for this includes the short SST training duration (13k steps) relative to pretraining (300k steps), and a loss function (scaled cosine similarity) that prioritizes smooth latent transitions over logical correctness.

3.  **Superior In-Context Scaling:** Unlike the Base Transformer, which stagnates between 5-shot and 8-shot tasks (showing only 0.1% improvement), JEPA-Reasoner effectively utilizes additional examples to significantly boost accuracy (improving from 37.1% to 48.2%).

**Comparison with models following other paradigm:**
| Paradigm | Example | Model Size | 8-shot Score (%) | Info Source |
| ------------- | ----------- | --------------- | -------- | -------- |
| Standard model | Gemma 3 | 4B | 38.4 | Official huggingface model card |
| CoT model | Llama 3.2 | 1B | 44.4 | Official huggingface model card |
| CoT model | Qwen 3 | 0.6B | 42.5 | lm-eval result (since official doc does not contain an 8-shot score) |
| Recurrent Depth | Huginn-0125 | 3.5B | 42.1 | Original paper |
| **Ours** | **JEPA-Reasoner** | **0.9B** | **48.2** | **lm-eval result** |

(As for COCONUT, we couldn't find an 8-shot GSM8k score. Considering that the authors of COCONUT has acknowledged that COCONUT performs worse than standard CoT as a trade off for efficiency in [their paper (Section 5.3 - Results and Discussion)](https://arxiv.org/html/2412.06769v3#:~:text=Although%20Coconut%20does%20not%20surpass%20CoT%20on%20GSM8k%2C%20it%20offers%20a%20superior%20trade%2Doff%20between%20reasoning%20efficiency%20and%20accuracy%20(Figure%208%2C%20I).), we did not conduct benchmarking for COCONUT models)

(lm-eval config was added to supplement)

We have updated paper with benchmark results and analysis. Refer to section 6 (Real World Evaluation) for more detail. We believe these benchmarks address the concerns regarding natural language performance. We are happy to answer any further questions.

---

### Author Response · Authors · 2025-11-29
**GSM8K Benchmarks (Update 1)**

# Regarding Natural Language Benchmarks

Here is the updated benchmark result (GSM8K) subsequent to [early results](https://openreview.net/forum?id=7ruA2rXG42&noteId=xpeAIpOnmC)

**Experimental Setup:**
- **Pretraining:** 300k steps
- **SST:** 25k-step snapshot    ⇦ Updated SST snapshot!
- **Talker:** 500 steps               ⇦ New Talker module for corresponding SST snapshot
- **Total Model Size:** 892M parameters (694M JEPA-Reasoner + 198M Mono-Talker)

**Comparison with models following other paradigm:**
| Paradigm | Example | Model Size | 8-shot Score (%) | Info Source |
| ------------- | ----------- | --------------- | -------- | -------- |
| Standard model | Gemma 3 | 4B | 38.4 | Official huggingface model card |
| CoT model | Llama 3.2 | 1B | 44.4 | Official huggingface model card |
| CoT model | Qwen 3 | 0.6B | 42.5 | lm-eval result (since official doc does not contain an 8-shot score) |
| Recurrent Depth | Huginn-0125 | 3.5B | 42.1 | Original paper |
| **Ours** | **JEPA-Reasoner** | **0.9B** | **51.7** | **lm-eval result** |

(As for COCONUT, we couldn't find an 8-shot GSM8k score. Considering that the authors of COCONUT has acknowledged that COCONUT performs worse than standard CoT as a trade off for efficiency in [their paper (Section 5.3 - Results and Discussion)](https://arxiv.org/html/2412.06769v3#:~:text=Although%20Coconut%20does%20not%20surpass%20CoT%20on%20GSM8k%2C%20it%20offers%20a%20superior%20trade%2Doff%20between%20reasoning%20efficiency%20and%20accuracy%20(Figure%208%2C%20I).), we did not conduct benchmarking for COCONUT models)

(lm-eval config was added to supplement)

With better benchmark comparing to [early results](https://openreview.net/forum?id=7ruA2rXG42&noteId=xpeAIpOnmC), we think it a strong evidence for JEPA-Reasoner's superior reasoning performance and NLP ability, while also further supporting the observation and conclusions made in [early results](https://openreview.net/forum?id=7ruA2rXG42&noteId=xpeAIpOnmC):

> Key Observations:
> 1. Significant Accuracy Gains: After adapting to the decoupled latent reasoning paradigm, performance increased by 79.2% in 5-shot and 131.7% in 8-shot settings compared to the baseline.
> 2. Architectural Efficiency: This improvement should be attribute to the decoupled latent space reasoning paradigm rather than the SST phase. Evidence for this includes the short SST training duration (13k steps) relative to pretraining (300k steps), and a loss function (scaled cosine similarity) that prioritizes smooth latent transitions over logical correctness.
> 3. Superior In-Context Scaling: Unlike the Base Transformer, which stagnates between 5-shot and 8-shot tasks (showing only 0.1% improvement), JEPA-Reasoner effectively utilizes additional examples to significantly boost accuracy (improving from 37.1% to 48.2%).

We have updated paper with benchmark results and analysis. Refer to section 6 (Real World Evaluation) for more detail. We believe these benchmarks address the concerns regarding natural language performance. We are happy to answer any further questions.

---

### Meta-Review · Area_Chair_u9pR · 2026-01-06

**Summary:**

Reviewer YehF: The major concerns influencing the suggested decision are limitations in evaluation scope and practical validation. The experimental results are confined to synthetic tasks, leaving open questions about whether the observed benefits transfer to natural language benchmarks or real-world NLP settings. In addition, the paper does not provide sufficient analysis of computational cost, training complexity, or inference efficiency. The absence of comparisons on standard benchmarks, unclear scalability characteristics, and lack of discussion around few-shot learning or prompting scenarios collectively reduce confidence in the method’s practical applicability and generalization.

Reviewer 6LZ2 considers the paper insufficiently developed for acceptance. The main concern is that the empirical evaluation relies almost entirely on two synthetic settings whose real-world relevance is unclear, particularly the use of Gaussian noise in latent space, and even in the more plausible CFG task the practical significance of the improvements is not well explained. The paper provides limited discussion and interpretation of results, making it difficult to understand why the findings matter in practice, and the related work on latent-space reasoning is incomplete, omitting recent relevant studies. In addition, the writing lacks clarity in several places, and the method appears to depend on specific architectural choices without ablation or analysis to justify them.

Reviewer Daki viewed the paper promising but incomplete: the architecture and synthetic results suggest potential, yet the objective-function mismatch, limited evaluation scope, and unclear theoretical justification for EMA in reasoning prevent him/her from recommending acceptance at this stage.

Reviewer 6HMd's primary concern is that the experimental evidence is too limited in scope and depth for a method claiming general improvements in reasoning. All evaluations are restricted to two synthetic tasks (tree search and CFG generation), with no results on standard reasoning benchmarks (e.g., GSM8K or similar). This makes it difficult to assess whether the proposed architecture offers practical advantages beyond carefully controlled settings. In addition, the reported robustness gains are modest and inconsistent. Lastly, the lack of ablation studies and insufficient baseline comparisons are also major limitations of the paper.

**Reviewer Concerns:**

Reviewer YehF: The rebuttal meaningfully addresses the core conceptual concerns regarding efficiency, inference overhead, and applicability beyond synthetic tasks. However, empirical validation of efficiency and scalability remains incomplete, and the new benchmark results, while encouraging, are not yet sufficient to fully resolve concerns about real-world performance and training practicality.

Reviewer 6LZ2: The rebuttal addresses several of the reviewer’s key concerns by clarifying the motivation and design of the two synthetic noise models, expanding the related work to include and distinguish recent latent-reasoning approaches, improving the exposition, and most importantly, adding GSM8K results that directly respond to the criticism of synthetic-only evaluation. However, some concerns remain at least partially outstanding: the real-world evaluation is still limited in breadth and based on early training snapshots, the practical significance of the gains may still feel preliminary, and no new ablation or empirical analysis is provided to justify the specific architectural design choices. As a result, while the rebuttal strengthens the paper and reduces ambiguity, it may not fully resolve concerns about the maturity and robustness of the contribution.

Reviewer Daki: The rebuttal successfully clarifies the design intent and motivation behind EMA-based latent learning and partially expands empirical validation. However, core concerns about objective-function alignment with reasoning correctness and the sufficiency of real-world evaluation remain partially unresolved.

Reviewer 6HMd: The rebuttal successfully addresses several technical and diagnostic concerns, particularly regarding model scaling behavior and architectural design choices. However, the core issues, including limited benchmark coverage, immature real-world evaluations, and narrow baseline comparisons, remain.

**Reviewer Scores:**

Reviewer YehF: The rebuttal would likely solidify the reviewer’s support rather than elevate it, leading to a maintained score of 6: marginally above the acceptance threshold.

Given the rebuttal, Reviewer 6LZ2 would likely increase the score modestly but remain negative overall. The added GSM8K results, clearer explanation of the noise models, expanded related work, and improved clarity directly address several major criticisms, which could plausibly move the reviewer from a 2 (reject) to a 4 (marginally below the acceptance threshold).

Reviewer Daki: Based on the rebuttal quality and reviewer’s original stance, the most likely score change is a modest increase, but not a full reversal.

Reviewer 6HMd would likely view the paper as clearly stronger than initially perceived but still not quite ready, leading to a maintained score of 4: marginally below the acceptance threshold.

Overall, the AC finds the paper to be borderline; while the ideas are promising, the evaluation is not yet sufficient to justify acceptance.

---

### Decision · Program_Chairs · 2026-01-26

Reject